# Prognostic Factors for Triple-Negative Breast Cancer with Residual Disease after Neoadjuvant Chemotherapy

**DOI:** 10.3390/jpm13020190

**Published:** 2023-01-21

**Authors:** Zhijun Li, Yiqun Han, Jiayu Wang, Binghe Xu

**Affiliations:** Department of Medical Oncology, National Cancer Center/National Clinical Research Center for Cancer/Cancer Hospital, Chinese Academy of Medical Sciences and Peking Union Medical College, Beijing 100021, China

**Keywords:** triple-negative breast neoplasms, non-pathologic complete response, *PIK3CA*, prognosis

## Abstract

Valid factors to evaluate the prognosis of triple-negative breast cancer (TNBC) with residual disease after neoadjuvant chemotherapy (NAC) are still lacking. We performed this study to explore prognostic factors focusing on genetic alterations and clinicopathology features in non- pathologic complete response (pCR) TNBC patients. Patients initially diagnosed with early-stage TNBC, treated with NAC, and who had residual disease after primary tumor surgery at the China National Cancer Center during 2016 and 2020 were enrolled. Genomic analyses were performed by targeted sequencing for each tumor sample. Univariable and multivariable analyses were conducted to screen prognostic factors for the survival of patients. Fifty-seven patients were included in our study. Genomic analyses showed that *TP53* (41/57, 72%), *PIK3CA* (12/57, 21%), and *MET* (7/57, 12%), and *PTEN* (7/57, 12%) alternations commonly occurred. The clinical TNM (cTNM) stage and *PIK3CA* status were independent prognostic factors of disease-free survival (DFS) (*p* < 0.001, *p* = 0.03). A prognostic stratification indicated that patients with clinical stages I &II possessed the best DFS, followed by those with clinical stage III & wild-type *PIK3CA*. In contrast, patients with clinical stage III & the *PIK3CA* mutation had the worst DFS. In TNBC patients with residual disease after NAC, prognostic stratification for DFS was observed by combining the cTNM stage and *PIK3CA* status.

## 1. Introduction

Breast cancer is one of the most common malignancies among women, with an estimated 2.3 million new cases each year worldwide [1]. Triple-negative breast cancer (TNBC) represents 15–20% of breast cancer cases and is associated with a lack of estrogen receptor (ER), progesterone receptor (PR), and human epidermal growth factor receptor-2 (HER2) expression [2]. A higher rate of recurrence and mortality is observed in TNBC than in other breast cancer subtypes, especially within the first three years [3]. At the same time, the mainstay therapeutic options are limited to surgery and chemotherapy due to the lack of these therapeutic targets [4,5]. 

The American Society of Clinical Oncology (ASCO) guidelines recommend neoadjuvant chemotherapy (NAC) regimens for TNBC patients with clinically node-positive or at least T1c disease to obtain surgical treatment opportunities and drug sensitivity information [6]. Approximately 25% to 45% of TNBC patients achieve a pathologic complete response (pCR) after standard anthracycline and taxane NAC [7,8,9]. The remaining patients with residual disease had a significantly worse prognosis and often experienced early recurrence or metastasis. Platinum has been confirmed to be an effective agent for TNBC to increase pCR or objective response rates, especially in those harboring the *BRCA1/2* mutation. However, its impact on disease-free survival (DFS) and overall survival (OS) is unclear [10,11,12,13,14,15]. Recently, the exploration of deep learning through convolutional neural networks (CNN) to predict NAC response in breast cancer has achieved good results and is expected to be widely used in clinical practice [16,17].

Inherent heterogeneity exists in TNBC tumors due to various gene somatic mutations. The genomic profile of tumors is shown to be frequently altered during chemotherapy. Several genetic markers that influence prognosis or predict efficacy have been reported prior. *TP53*, the most frequent mutant gene in TNBC, has been confirmed to be a predictor of chemotherapy resistance but not an effective prognostic marker [18]. Approximately ~15% of TNBC patients harbor a *BRCA1/2* germ-line mutation. The products BRCA1/2 proteins were vital in DNA double-strand break repair by homologous recombination [19,20]. The POSH study showed that patients with a *BRCA1/2* germ-line mutation had higher OS than those with wild-type *BRCA1/2*, while the results might be related to the greater sensitivity of *BRCA1/2* germ-line carriers to chemotherapy [21]. 

Valid factors to evaluate prognosis for TNBC patients with residual disease after NAC are still lacking. This study focuses on genetic alterations and clinicopathology features in non-pCR TNBC patients after NAC to explore influential prognostic factors which will guide clinical practice and promote scientific research.

## 2. Materials and Methods

### 2.1. Study Population

Patients diagnosed with early-stage TNBC, treated with NAC, and who had residual disease after primary tumor surgery at the China National Cancer Center between January 2016 and December 2020 were enrolled from a total of 2614 patients. Patients who met the following criteria were included in the study: (1) female patients aged 18–80 years; (2) clinical stage I~III breast cancer at initial diagnosis; (3) TNBC: histologically confirmed breast cancer with estrogen-receptor-negative, progesterone-receptor-negative, and HER2-negative breast cancer; (4) NAC with anthracycline plus paclitaxel (AP) or platinum plus paclitaxel (PP) for 6~8 cycles; (5) surgical resection for the primary tumor; and (6) a residual tumor in postoperative pathology and a sufficient tissue sample that could be obtained for genetic testing (at least eight sections of FFPE tumor tissue and surrounding normal tissue with a thickness of 4–5 µm). Patients with primary bilateral breast cancer or other malignancies were excluded.

The study was reviewed and approved by the Institutional Review Board of the National Cancer Center/Cancer Hospital, Chinese Academy of Medical Sciences and Peking Union Medical College in May 2021 (approval number: 21/247-2918). All of the procedures were carried out in accordance with the Declaration of Helsinki.

### 2.2. DNA Extraction and Capture-Based Targeted DNA Sequencing

Genomic DNA was extracted from formalin-fixed paraffin-embedded (FFPE) tumor tissues with at least 10% tumor content using a QIAamp DNA FFPE tissue kit (Qiagen, Hilden, Germany). Fragments between 200–400 bp from the sheared tissue DNA were purified (Agencourt AMPure XP Kit, Beckman Coulter, Pasadena, CA, USA), hybridized with capture probe baits, selected with magnetic beads, and amplified. Target capture of the tissue DNA samples was performed using a 520-gene panel spanning 1.64 megabases of the human genome. The quality and the size of the fragments were assessed by a high-sensitivity DNA kit using a Bioanalyzer 2100 (Agilent Technologies, Santa Clara, CA, USA). Indexed samples were sequenced on Nextseq 500 (Illumina, Inc., San Diego, CA, USA) with paired-end reads and an average sequencing depth of 1000× for the tissue samples.

### 2.3. Sequence Data Analysis

Sequence data were mapped to the reference human genome (hg19) using Burrows–Wheeler Aligner version 0.7.10 [22]. Local alignment optimization, duplication marking, and variant calling were performed using Genome Analysis Tool Kit version 3.2 [23] and VarScan version 2.4.3 [24]. To identify somatic variants, tissue samples were compared against their own white blood cell control. Variants were filtered using the VarScan fpfilter pipeline, and loci with depths less than 100 were filtered out. Variants with a population frequency over 0.1% in the ExAC, 1000 Genomes, dbSNP, or ESP6500SI-V2 databases were grouped as single nucleotide polymorphisms and excluded from further analysis. The remaining variants were annotated with ANNOVAR (1 February 2016 release) [25] and SnpEff version 3.6 [26]. Analysis of structural variations was performed using FACTERA version 1.4.3 [27]. Copy number variations (CNVs) were analyzed based on the depth of coverage data of capture intervals. The copy number was calculated based on the ratio between the depth of the coverage in tumor samples and the average range of an adequate number (n > 50) of samples without CNVs as references per capture interval. CNV is called if the coverage data of the gene region is quantitatively and statistically significant from its reference control. The detection limit for CNVs is 1.5 for copy number deletion and 2.64 for copy number amplifications.

### 2.4. Statistical Analysis

DFS was defined as the time from primary tumor surgery to the first event of either recurrence or death. Clinicopathological features of patients were summarized based on NAC regimens, and differences in category variables were compared using the Wilcoxon test or Fisher’s exact test. The cumulative DFS was estimated using the Kaplan–Meier analysis and compared between groups using the log-rank test. Univariable and multivariable cox regression analyses were conducted to screen potential prognostic factors of the patients. *p*-values < 0.05 were considered statistically significant. Statistical analysis was performed using R version 4.0.3.

## 3. Results

### 3.1. Baseline Characteristics of the Overall Cohort and Treatment Subgroups

The flow chart used to screen eligible patients is presented in Figure 1. Fifty-seven non-PCR TNBC patients treated with NAC and primary tumor surgery were screened for inclusion in this study from 2614 patients with early breast cancer. The median age of general patients was 45 (ranging from 38 to 52). Moreover, the majority of the cohort comprised patients with clinical stage II and III breast cancer (56/57, 98.2%), and only one patient had clinical stage I disease. The cohort was further grouped based on NAC regimens and divided into the AP group (33/57, 57.9%) and the PP group (24/57, 42.1%). No significant difference between the subgroups for all of the variables was observed. Patient characteristics at the baseline of the overall cohort and treatment subgroups are summarized in Table 1.

### 3.2. Alteration Landscape of the Cohort

The alteration profiles of 57 breast tumor samples were analyzed by next-generation sequencing (NGS) using a panel of 520 cancer-related genes and are shown in Figure 2. *TP53* presented the highest alteration frequency, accounting for approximately 72% (41/57 patients), followed by *PIK3CA* (12/57 patients, 21%), *MET* (7/57 patients, 12%), and *PTEN* (7/57 patients, 12%). Tumors with *PTEN* and *NF1* alteration often occurred in patients with pathologic stage II & III after NAC. Seven tumors with *PTEN* alteration and four with *NF1* alteration were presented in patients with pathologic stage II & III, while none of the patients with pathologic stage I had these two gene alterations.

### 3.3. Treatment Outcomes

We analyzed the clinical and pathology treatment outcomes of 57 non-pCR TNBC patients who received NAC (Table 1, Appendix A). A value of 68.4% of patients had significant tumor shrinkage with Miller–Payne grades 3–4, whereas 31.6% were found to possess minor tumor shrinkage with Miller–Payne grades 1–2, which was evaluated by histopathology. Premenopausal patients had a higher proportion of Miller–Payne grades 3–4 in the PP group (*p* = 0.02). A value of 84% of premenopausal patients (21/25) and 37.5% of menopausal patients (3/8) had Miller–Payne grades 3–4, respectively. In contrast, the proportion of Miller–Payne grades 3–4 was higher in menopausal patients in the AP group (*p* = 0.022). A value of 47.1% of premenopausal patients (8/17) had Miller–Payne grades 3–4, compared with 100% (7/7) of the menopausal patients (Appendix A).

Regarding the relationship between molecular features and treatment outcomes, we found that patients harboring the *BRCA1/2* mutation were insensitive to chemotherapy, especially anthracycline agents (Appendix A). Tumors with wild-type *TP53* had more sensitivity to chemotherapy than those with the *TP53* mutation (*p* = 0.011), regardless of treatment regimens (Appendix A). Miller–Payne grades 3–4 were obtained in 93.8% (15/16) of tumor samples with wild-type *TP53*, compared to 58.5% (24/41) of tumor samples with mutant *TP53*.

### 3.4. Clinical Survival Outcomes

The median follow-up time was 37.1 months (95% CI 33.1–41.1) for the cohort. Nineteen (33.3%) patients developed recurrence or metastasis as of March 2022.

#### 3.4.1. Clinicopathology Features and DFS

Of the 57 non-pCR TNBC patients, NAC with PP resulted in a significantly improved DFS than AP (*p* = 0.047) (Figure 3A). The three-year DFS rate was 54.5% compared with 74.5%, and the median DFS was 45.7 and not reached for the AP group and the PP group, respectively. In addition, patients with clinical stage III disease had a significantly shorter DFS than patients with clinical stage I and II disease (HR 9.28, 95% CI 2.13–40.36, *p* < 0.001) (Figure 3B). The clinical N (cN) stage had the most significant impact on prognosis, among which patients with the cN0 stage had the longest DFS, while patients with the cN3 stage had the shortest DFS (*p* < 0.001) (Figure 3C). Furthermore, tumors with lymph vessel invasion resulted in worse DFS in patients than non-lymph vessel invasion (HR 2.66, 95% CI 0.92–7.69, *p* = 0.061) (Figure 3D). However, whether adjuvant chemotherapy or radiotherapy was performed did not affect DFS. (Appendix A).

Subgroup analysis of DFS was performed for patients based on the treatment subgroups. The results showed that patients with clinical stage I and II disease had a significantly longer DFS than those with clinical stage III disease both in the PP group (*p* = 0.004) and the AP group (*p* = 0.01). (Appendix A).

#### 3.4.2. Molecular Features and DFS

The association of molecular features and DFS in TNBC was analyzed (Table 2; Appendix A). There were no apparent correlations between *TP53* and DFS (Appendix A). Patients harboring the *PIK3CA* mutation showed worse DFS than patients with wild-type *PIK3CA* (HR 2.75, 95% CI 1.07–7.03, *p* = 0.028) (Figure 3E; Appendix A). The three-year DFS rate was 40.0% compared with 72.3%, and the median DFS was 23.8 and not reached for the mutation and wild-type groups, respectively. However, this difference was more pronounced in patients receiving platinum-containing NAC (HR 4.44, 95% CI 1.05–18.76, *p* = 0.026) but not in patients receiving anthracycline-containing NAC (HR 1.33, 95% CI 0.38–4.62, *p* = 0.649). Patients with mutant *MYC* tended to respond better to PP chemotherapy than those with wild-type *MYC* (Appendix A). Disappointingly, no significant correlations between molecular features and metastatic sites were found.

#### 3.4.3. Prognostic Risk Factors and Stratified Analysis

Univariate analysis of clinical characteristics and molecular features in non-pCR TNBC patients was performed. It has been shown that the clinical TNM (cTNM) stage, cN stage, pathologic N (pN) stage, *PIK3CA* mutation status, and NAC regimens were associated with DFS (*p* < 0.05). Multivariable Cox analysis confirmed that cTNM stage and *PIK3CA* mutation status were highly significant independent prognostic factors of DFS in non-pCR TNBC patients (*p* < 0.05) (Table 2).

We carried out a prognostic risk assessment in non-pCR TNBC patients by combining these two independent prognostic factors (Figure 4). The results showed that patients with clinical stage I and II had the best DFS (a three-year DFS rate of 96%), followed by those with clinical stage III & wild-type *PIK3CA* (a three-year DFS rate of 53.7%), while those with clinical stage III & *PIK3CA* mutation possessed the worst DFS (a three-year DFS rate of 0) (*p* < 0.001). Therefore, the flow diagram shown in Figure 4 could be a good stratification of DFS for non-pCR TNBC patients.

## 4. Discussion

In patients with non-pCR TNBC after NAC, the DFS in the PP group was longer than that in the AP group. The three-year DFS rate was 54.5% in the AP group compared with 74.5% in the PP group. Several studies have shown that TNBC patients receiving platinum-containing NAC achieved a higher pCR rate than those without platinum regimens. In the GeparSixto trial, 315 patients of stage II & III TNBC were assigned to receive paclitaxel + liposomal doxorubicin + bevacizumab with or without carboplatin as neoadjuvant treatment. The pCR rate was increased from 36.9% to 53.2% after adding carboplatin to NAC (*p* = 0.005) [13]. While event-free survival (EFS) and OS have been controversial in prior research, studies have indicated similar, even worse results in platinum-containing regimens than platinum-free regimens of NAC [8,28]. However, the differences in the above survival results cannot rule out the strong inherent heterogeneity in TNBC. Therefore, we tried to explore prognostic differences at the gene signatures level with a view to finding effective prognostic factors.

Our data demonstrated divergent sensitivity to different chemotherapeutic agents when patients were grouped according to their menopausal status. Premenopausal patients were sensitive to PP treatment, while menopausal patients responded well to AP treatment. Our prior study showed that premenopausal patients were more likely to have homologous recombination deficiency (HRD). As we know, patients with HRD are more responsive to platinum-containing therapy [29]. Therefore, the relationship between menopausal status and chemotherapy sensitivity might primarily be attributable to HRD. Moreover, we observed that all patients with *PTEN* or *NF1* alteration remained in pathologic stages II and III despite receiving NAC. As previously reported, patients who possess *NF1* mutations or lose the expression of *PTEN* have worse survival in breast cancer [30,31].

In our study, the *PIK3CA* mutation frequency in TNBC patients accounted for 15%, which is consistent with previous reports [32]. Our results showed that *PIK3CA* mutation might be a poor prognostic indicator for DFS. Patients harboring the *PIK3CA* mutation had a worse three-year DFS rate than those with wild-type *PIK3CA*, for 40.0% and 72.3% of patients, respectively (*p* = 0.028). Similarly, previous studies have confirmed that the *PIK3CA* mutation confers resistance to chemotherapy in TNBC by inhibiting apoptosis and activating the PI3K/AKT/mTOR signaling pathway [33], and TNBC patients with the *PIK3CA* mutation had lower pCR rates than those with wide-type *PIK3CA* after NAC [34]. Interestingly, we found that the difference in DFS was more pronounced in patients receiving PP NAC (*p* = 0.026), while no significant difference existed among those undergoing AP treatment. Prior research concluded no predictive effect of *PIK3CA* mutation for doxorubicin treatment and whether harboring the *PIK3CA* mutation cannot affect the DFS or OS [35]. However, no studies have reported the impact of platinum agents on DFS in *PIK3CA*-mutated TNBC patients. Therefore, further research is needed on the mechanism of action and prognostic effect of different chemotherapeutic agents on *PIK3CA*-mutated TNBC.

It was well known that the TNM stage is closely related to prognosis. In this study, cTNM stage and *PIK3CA* mutation were confirmed to be highly significant independent prognostic factors of DFS in non-pCR TNBC patients. We performed a prognostic risk assessment using cTNM stage and *PIK3CA* mutation status in non-pCR TNBC patients according to the univariate and multivariable Cox analysis of clinical characteristics and molecular features. A better prognostic stratification for patients was observed. We found that patients with clinical stages I & II possessed the best DFS, while those with clinical stage III & the *PIK3CA* mutation had the worst DFS. Therefore, we can further evaluate the prognosis for non-pCR TNBC patients by combining cTNM stage and *PIK3CA* mutation status.

To our knowledge, this is the first study to explore prognostic factors for patients with non-pCR TNBC by combining clinicopathology and molecular features. However, our study does have several limitations. First, this was a small sample cohort, and targeted sequencing could not be performed in some samples due to insufficient tumor samples. Second, joint analysis of multiple genes could not be performed due to the low incidence of gene mutations. Third, analyses of therapeutic agents such as PI3K inhibitors and PARP inhibitors are lacking.

## 5. Conclusions

In TNBC patients with residual disease after NAC, a promising prognostic stratification was observed by combining the cTNM stage and *PIK3CA* mutation status. These findings need to be further verified.

## Figures and Tables

**Figure 1 jpm-13-00190-f001:**
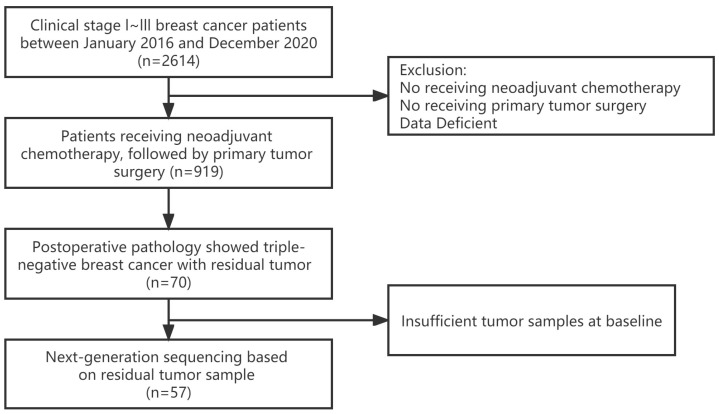
Flowchart of the screening procedure.

**Figure 2 jpm-13-00190-f002:**
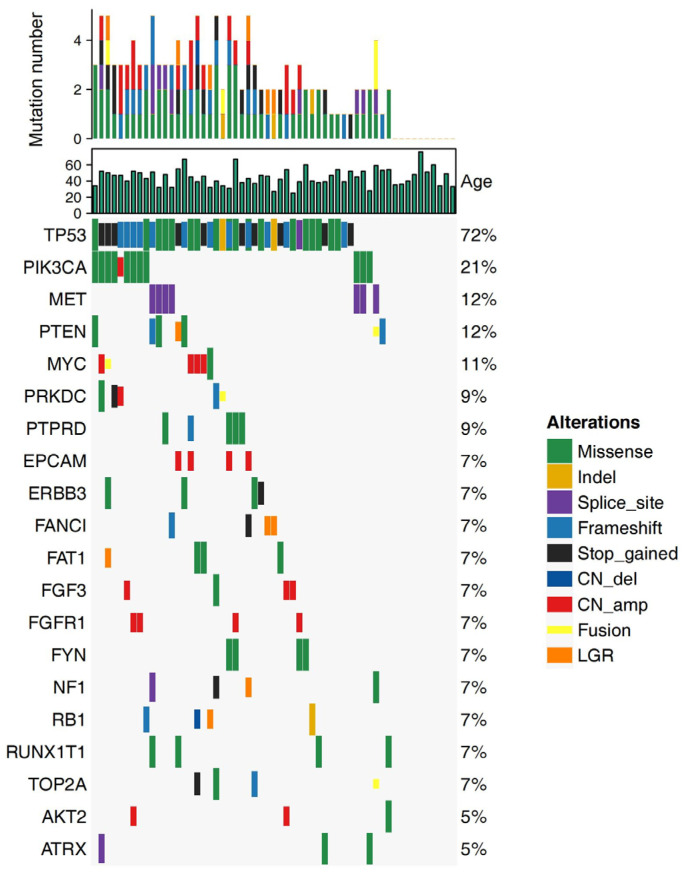
Alteration profiles of the top 20 genes from 57 patients with non-pCR TNBC. pCR, pathologic complete response; TNBC, triple-negative breast cancer.

**Figure 3 jpm-13-00190-f003:**
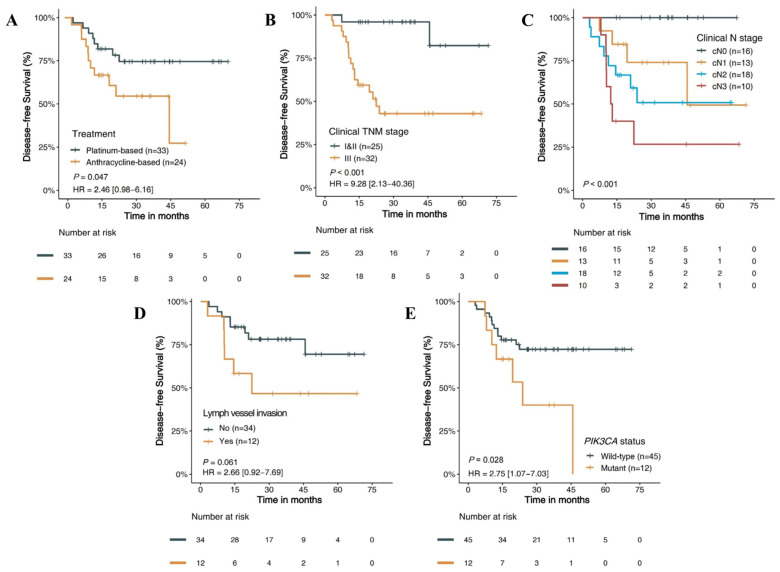
Kaplan–Meier survival curves for DFS of patients with non-pCR TNBC. DFS based on (**A**) NAC regimens; (**B**) clinical TNM stage; (**C**) clinical N stage; (**D**) lymph vessel invasion; (**E**) *PIK3CA* status. DFS, disease-free survival; pCR, pathologic complete response; TNBC, triple-negative breast cancer; NAC, neoadjuvant chemotherapy; cTNM, clinical TNM stage; cN, clinical N stage; HR, hazard ratio.

**Figure 4 jpm-13-00190-f004:**
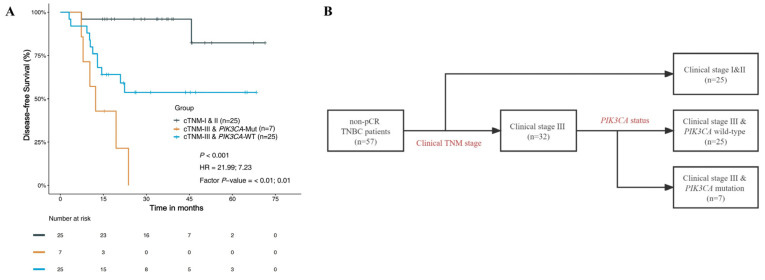
Prognostic risk assessment based on clinical and molecular features of patients with non-pCR TNBC. (**A**) DFS based on clinical TNM stage and *PIK3CA* status; (**B**) flowchart of prognostic risk assessment. DFS, disease-free survival; pCR, pathologic complete response; TNBC, triple-negative breast cancer; cTNM, clinical TNM stage; Mut, mutation; WT, wild-type; HR, hazard ratio.

**Table 1 jpm-13-00190-t001:** Baseline characteristics and treatment outcomes of the overall cohort and treatment subgroups.

Variable	Overall (n = 57)	Platinum–Paclitaxel Regimen (n = 33)	Anthracycline–Paclitaxel Regimen (n = 24)	*p*-Value
Age				0.312
Median [range]	45.0 [38.0, 52.0]	43.0 [36.0, 51.0]	46.5 [38.0, 54.0]	
Menopausal status				0.765
Premenopause	42 (73.7%)	25 (75.8%)	17 (70.8%)	
Postmenopause	15 (26.3%)	8 (24.2%)	7 (29.2%)	
Family history of cancer				0.584
No	38 (66.7%)	23 (69.7%)	15 (62.5%)	
Yes	19 (33.3%)	10 (30.3%)	9 (37.5%)	
Ki-67(%)				0.416
Median [range]	55.0 [21.3, 70.0]	50.0 [10.0, 70.0]	60.0 [30.0, 75.0]	
Missing	1 (1.8%)	0 (0.0%)	1 (4.2%)	
Clinical T stage				0.417
cT1	6 (10.5%)	2 (6.1%)	4 (16.7%)	
cT2	36 (63.2%)	23 (69.7%)	13 (54.2%)	
cT3	12 (21.1%)	7 (21.2%)	5 (20.8%)	
cT4	3 (5.3%)	1 (3.0%)	2 (8.3%)	
Clinical N stage				0.836
cN0	16 (28.1%)	9 (27.3%)	7 (29.2%)	
cN1	13 (22.8%)	9 (27.3%)	4 (16.7%)	
cN2	18 (31.6%)	10 (30.3%)	8 (33.3%)	
cN3	10 (17.5%)	5 (15.2%)	5 (20.8%)	
Clinical TNM stage				0.794
I and II	25 (43.9%)	15 (45.5%)	10 (41.7%)	
III	32 (56.4%)	18 (54.5%)	14 (58.3%)	
Pathologic T stage				1.000
ypT1	39 (68.4%)	23 (69.7%)	16 (66.7%)	
ypT2	18 (31.6%)	10 (30.3%)	8 (33.3%)	
Pathologic N stage				0.497
ypN0	26 (45.6%)	14 (42.4%)	12 (50.0%)	
ypN1	18 (31.6%)	13 (39.4%)	5 (20.8%)	
ypN2	6 (10.5%)	3 (9.1%)	3 (12.5%)	
ypN3	7 (12.3%)	3 (9.1%)	4 (16.7%)	
Pathologic TNM stage				0.376
I	21 (36.8%)	11 (33.3%)	10 (41.7%)	
II	23 (40.4%)	16 (48.5%)	7 (29.2%)	
III	13 (22.8%)	6 (18.2%)	7 (29.2%)	
Miller–Payne				0.565
G1 and G2	18 (31.6%)	9 (27.3%)	9 (37.5%)	
G3 and G4	39 (68.4%)	24 (72.7%)	15 (62.5%)	
Histological grade				0.732
Grade II	11 (19.30%)	7 (21.21%)	4 (16.67%)	
Grade III	35 (61.40%)	19 (57.58%)	16 (66.67%)	
Missing	11 (19.30%)	7 (21.21%)	4 (16.67%)	
Lymph vessel invasion				0.738
No	34 (59.7%)	20 (60.6%)	14 (58.3%)	
Yes	12 (21.1%)	6 (18.2%)	6 (25.0%)	
Missing	11 (19.3%)	7 (21.2%)	4 (16.7%)	
Adjuvant chemotherapy				0.787
No	24 (42.1%)	13 (39.4%)	11 (45.8%)	
Yes	33 (57.9%)	20 (60.6%)	13 (54.2%)	
Adjuvant radiotherapy				1.000
No	9 (15.8%)	5 (15.2%)	4 (16.7%)	
Yes	48 (84.2%)	28 (84.9%)	20 (83.3%)	
Initial metastatic sites				0.168
Bone/soft tissue only	12 (21.1%)	6 (18.2%)	6 (25.0%)	
Visceral	7 (12.3%)	2 (6.1%)	5 (20.8%)	
None	38 (66.7%)	25 (75.8%)	13 (54.2%)	
Number of metastatic sites				1.000
1	12 (21.1%)	5 (15.2%)	7 (29.2%)	
2	7 (12.3%)	3 (9.1%)	4 (16.7%)	
None	38 (66.7%)	25 (75.8%)	13 (54.2%)	
*BRCA1/2* status				0.073
Mutant	6 (10.5%)	1 (3.0%)	5 (20.8%)	
Wild-type	51 (89.5%)	32 (97.0%)	19 (79.2%)	

cT, clinical T; cN, clinical N; pT, pathologic T; pN, pathologic N.

**Table 2 jpm-13-00190-t002:** Univariate and multivariate survival analysis of 57 non-pathologic complete response triple-negative breast cancer patients.

Variable	Univariate		Multivariate	
	HR (95% CI)	*p*-Value	HR (95% CI)	*p*-Value
Treatment		0.047		
Platinum–Paclitaxel (n = 33)	Ref.			
Anthracycline–Paclitaxel (n = 24)	2.46 (0.98–6.16)			
Clinical N stage		<0.001		
cN0 and cN1 (n = 29)	Ref.			
cN2 (n = 18)	4.41 (1.32–14.72)			
cN3 (n = 10)	8.07 (2.33–27.98)			
Clinical TNM stage		<0.001		<0.001
I and II (n = 25)	Ref.		Ref.	
III (n = 32)	9.28 (2.13–40.36)		11.02 (3.02–40.2)	
Pathologic N stage		<0.001		
pN0& and N1 (n = 44)	Ref.			
pN2 (n = 6)	0.68 (0.09–5.25)			
pN3 (n = 7)	8.58 (2.78–26.51)			
*PIK3CA* status		0.028		0.03
Wild-type (n = 45)	Ref.		Ref.	
Mutant (n = 12)	2.75 (1.07–7.03)		2.70 (1.10–6.6)	

cN clinical N; pN pathologic N.

## Data Availability

The data used during the current study are available from the corresponding author on reasonable request.

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
