# Peer review of "Prognostic Factors for Triple-Negative Breast Cancer with Residual Disease after Neoadjuvant Chemotherapy"

_jpm, 2023, doi:10.3390/jpm13020190_

Round 1

Reviewer 1 Report

The authors evaluated the prognosis of triple-negative breast cancer with residual disease after neoadjuvant chemotherapy using 520-gene panel containing known cancer genes. The analysis was performed on cancer tissue samples. Two treatment regimens were applied to the patients. the authors provide genetic alteration profiles of the top 20 genes from the patients under study. They performed statistical analysis including molecular markers and clinical course of disease. They show that the addition of carboplatin to the chemotherapy increased the patients’ survival.

The authors also stress the limitations they had. TNBC cancer represents about 15% of all breast cancers, it is the most aggressive type of breast cancer and there is no targeted treatment for it now. The No of patients is small, so the results of statistical analysis are not fully reliable. Nevertheless, the manuscript is interesting, the analysis is state of art.

Author Response

Dear Reviewer:

On behalf of my co-authors, we are very grateful to you for giving us an opportunity to revise our manuscript. We appreciate you very much for your positive and constructive comments and suggestions on our manuscript entitled “Prognostic factors for triple-negative breast cancer with residual disease after neoadjuvant chemotherapy” (jpm-2157288).

Thank you very much for your affirmation! Fifty-seven non-PCR TNBC patients treated with NAC and primary tumor surgery at China National Cancer Center between January 2016 and December 2020 were enrolled in our study. Due to the low incidence of triple-negative breast cancer, the relatively small number of non-pCR patients undergoing neoadjuvant chemotherapy, and to ensure the accuracy of the test results, we only included surgical specimens from the past 5 years, so the total population is relatively small. We look forward to accumulating more cases and tumor samples to verify our results further.

Thank you very much again!

Reviewer 2 Report

I’d kindly ask the Authors to answer some comments listed below. 

Author Response

Response to Reviewer 1 Comments

Dear Reviewer:

On behalf of my co-authors, we are very grateful to you for giving us an opportunity to revise our manuscript. we appreciate you very much for your positive and constructive comments and suggestions on our manuscript entitled “Prognostic factors for triple-negative breast cancer with residual disease after neoadjuvant chemotherapy” (jpm-2157288).

We have studied the comments carefully and tried our best to revise our manuscript according to the comments. The followings are the responses and revisions made in response to the questions and suggestions on an item-by-item basis. Thank you very much again!

Point 1: In the Introduction Authors introduce the Triple‐negative breast cancer and the neoadjuvant therapy as a recommended treatment to reach pathological complete response. Predicting this in neoadjuvant patients is not easy but CAD models, involving radiomics via Convolutional Neural Networks (CNN), have helped in this task with good performances. Therefore, in the Introduction, Authors could enrich the discussion of response to neoadjuvant therapy mentioning, for example, these state‐of‐the‐art works: 

Response 1: Thank you for your valuable suggestions and recommended literature! Convolutional Neural Networks are indeed a promising way to predict NAC response for breast cancer, and we've added the content to the introduction.

Point 2: I think that the plots in Figures 2), 3) and 4) are a bit small and reading the letters and the numbers is not easy. I understand the necessity to fit them inside the manuscript but could the Authors make them a bit bigger, if possible? Also Tables 1 and 2 seem to be differently placed in the manuscript. Table 1 is more shifted to right, instead, Table 2 is more shifted to the left. I’d suggest a more uniform style throughout the manuscript

Response 2: We thank the reviewer for pointing this out. We've re-edited the figures (2,3,4) and adjusted the table (1,2) positions uniformly.

Point 3: In line 124 Authors say “...divided into AP group...and the PP group...”. I guess these acronyms refer to Anthracycline‐ Paclitaxel and Platinum Paclitaxel subgroups of Table 1. However this is not immediate in the the text and these acronyms have not been previously mentioned in the text. I’d suggest the Authors to briefly introduce these acronyms for the reader at the beginning of the manuscript.

Response 3: We are very sorry for our negligence in this issue, and we have already supplemented it in the Materials and Methods section of the manuscript.

Point 4: In the Discussion section Authors mention various works of the literature. If possible could the Authors compare and summarize their results with the past literature?

Response 4: We appreciate your suggestion. We think your suggestion is very meaningful, and we have compared and discussed similar results that have been available in the discussion section. At the same time, we also analyzed the first findings of our study based on previous studies.

Point 5: In lines 285‐286 Authors mention the limitations of their work: small dataset, lack of multiple genes analysis, PI3K and PARP inhibitors. How do these elements combined could affect the results shown in the Discussion?

Response 5: Thank you for your question! First, if we have a large sample size in the future, the results will be more persuasive; Second, due to the low frequency of gene mutations, fewer mutated genes can be analyzed; If the sample size can be expanded, the number of gene mutations is relatively increased, then we can perform a variety of analysis, including multiple gene analysis, such as PAM pathway; Third, if patients are treated with PI3K or PARP inhibitors, we can analyze more prognostic factors and perform more subgroup analyses.

Point 6: In lines 284‐285 Authors say “... this is the first study to explore the prognostic factors for patients with non‐ pCR TNBC ...”. Therefore studies about this topic, using CAD methods, are lacking. I think that this topic could benefit from CAD methods involving Machine Learning and radiomic Deep Learning techniques. Indeed these methods have been used, through the years, in various breast cancer studies focusing sentinel lymph‐node, 1mammograms, pathologic complete response to neoadjuvant chemotherapy, etc. In this sense, in the Discussion, Authors could better discuss these CAD models and their advantages in the clinical practice taking into account, for example, these works:

Response 6: Thank you very much for your suggestions! We know that Machine learning can extract high-throughput information from MR images to reflect tumor heterogeneity and predict tumor response early in NAC or even before treatment. Deep learning through convolutional neural networks to predict NAC response in breast cancer has achieved good results and is expected to be widely used in clinical practice. However, our research mainly explores the prognostic factors of non-pCR breast cancer patients, including clinicopathology and gene mutation profiles. In the future, we will have the opportunity to explore in depth the efficacy prediction methods of breast cancer patients, focusing on these innovative and promising CAD methods.

Reviewer 3 Report

The manuscript is of practical value, even that gene profiling is still not widely possible to perform.

The importance of PIK3CA mutations in TNBC cancer patients is explored in metastatic setting, but this research suggest that PIK3CA targeted therapy could be studied as adjuvant therapy for TNBC patient not achieving pCR after NAC.

Author Response

Dear Reviewer:

On behalf of my co-authors, we are very grateful to you for giving us an opportunity to revise our manuscript. we appreciate you very much for your affirmation on our manuscript entitled “Prognostic factors for triple-negative breast cancer with residual disease after neoadjuvant chemotherapy” (jpm-2157288).

Thank you very much again!